# Preparation, Characterization and Multiple Biological Properties of Peptide-Modified Cerium Oxide Nanoparticles

**DOI:** 10.3390/biom12091277

**Published:** 2022-09-10

**Authors:** Mengjun Wang, Hongliang He, Di Liu, Ming Ma, Yu Zhang

**Affiliations:** State Key Laboratory of Bioelectronics, Jiangsu Key Laboratory for Biomaterials and Devices, School of Biological Sciences and Medical Engineering, Southeast University, Nanjing 210096, China

**Keywords:** cerium oxide nanoparticles, CeO_2_@PAA, CeO_2_@PAA@RGD, antioxidant, anti-inflammatory, angiogenesis

## Abstract

Although cerium oxide nanoparticles are attracting much attention in the biomedical field due to their unique physicochemical and biological functions, the cerium oxide nanoparticles greatly suffer from several unmet physicochemical challenges, including loss of enzymatic activity during the storage, non-specific cellular uptake, off-target toxicities, etc. Herein, in order to improve the targeting property of cerium oxide nanoparticles, we first modified cerium oxide nanoparticles (CeO_2_) with polyacrylic acid (PAA) and then conjugated with an endothelium-targeting peptide glycine-arginine-aspartic acid (cRGD) to construct CeO_2_@PAA@RGD. The physiochemical characterization results showed that the surface modifications did not impact the intrinsic enzymatic properties of CeO_2_, including catalase-like (CAT) and superoxide dismutase-like (SOD) activities. Moreover, the cellular assay data showed that CeO_2_@PAA@RGD exhibited a good biocompatibility and a higher cellular uptake due to the presence of RGD targeting peptide on its surface. CeO_2_@PAA@RGD effectively scavenged reactive oxygen species (ROS) to protect cells from oxidative-stress-induced damage. Additionally, it was found that the CeO_2_@PAA@RGD converted the phenotype of macrophages from proinflammatory (M1) to anti-inflammatory (M2) phenotype, inhibiting the occurrence of inflammation. Furthermore, the CeO_2_@PAA@RGD also promoted endothelial cell-mediated migration and angiogenesis. Collectively, our results successfully demonstrate the promising application of CeO_2_@PAA@RGD in the future biomedical field.

## 1. Introduction

Cerium oxide nanoparticles are rare-earth-metal-based nanoparticles with a characteristic fluorite structure [1,2]. The reversible conversion of Ce^3+^ and Ce^4+^ on the surface of cerium oxide nanoparticles endows them with good redox properties and self-regenerative properties. Due to the unique nanostructure and electronic properties, cerium oxide nanoparticles can be widely used in various fields such as biosensors [3], combustion aids [4], etc. Recently, it was reported that cerium oxide nanoparticles can mimic a variety of enzymatic activities, including oxidase [5,6], catalase [7], peroxidase and superoxide dismutase (SOD) [8], and even phosphatase activities [9], which allows it to have various therapeutic applications for various oxidative-stress-related diseases, including chronic inflammation [10], diabetes [11,12,13], neurology [14,15], lung injury [16], liver disease [17,18], cardiac hypertrophy [19], and cancer [20].

Despite much recent progress made in the biomedical application of cerium oxide nanoparticles, the further biomedical application of cerium oxide nanoparticles is severely hindered by some unmet physicochemical challenges, including loss of enzymatic activity during the storage, non-specific cellular uptake, off-target toxicities, etc. Among them, smaller cerium oxide particles (less than 5 nm) have been shown to have higher enzymatic activity due to their larger specific surface area [21,22]. However, the tendency to form aggregates in biological fluids decreases their specific surface area which reduces their enzymatic activity [5]. Moreover, the interaction of nanoparticles with the cell culture medium leads to irreversible aggregation of nanoparticles, which significantly increases their overall cytotoxicity [23]. It has been noted that nanoparticles without targeted modifications lead to insufficient uptake by cells, which greatly limits their bioavailability in vivo [24]. Therefore, in order to further advance the biological applications of cerium oxide nanoparticles, it is highly urgent to address the aggregation-induced stability issue and improve the targeting ability.

The surface modification of nanoparticles using some natural or synthetic biodegradable polymers can not only improve the biocompatibility of the particles, but also bestow the nanoparticles with additional functions resulting from the changed surface properties [5]. Some researchers have reported that surface modification with alginate and chitosan [25], polyvinyl pyrrolidone (PVP) [26], polyethylene glycol (PEG) [27], dextran [28], or polyacrylic acid (PAA) [29] greatly improves the enzyme activity and stability of cerium oxide nanoparticles in water. In addition, peptides are widely used for surface modification of nanomaterials to enhance the targeting ability, improve the biocompatibility, and bestow the nanoparticles with some peptide-associated biological functions [30,31]. Among them, arginine-glycine-aspartate (RGD) peptide was demonstrated to possess various biological properties, including promoting the attachment, diffusion, and differentiation of some cells into various lineages [32], and can bind effectively to the cells mediated by its cellular receptors integrins [33]. Hereafter, the RGD peptide has been widely used to modify nanoparticles to improve the targeting property and synergistic efficacy [34,35,36,37]. Despite many strategies made to address the limited enzymatic activity of cerium oxide nanoparticles due to the instability issue [6], the biological functions after a modification have rarely been explored.

With these in mind, the overall aim of this project is to improve both physiochemical and biological properties of cerium oxide nanoparticles with surface modifications. To be specific, PAA-modified cerium oxide nanoparticles (CeO_2_@PAA) with a negative surface charge and homogeneous particle size were first obtained by the precipitation method. Subsequently, CeO_2_@PAA was further conjugated with RGD peptides to construct the CeO_2_@PAA@RGD. Next, we thoroughly characterized the physiochemical properties of different cerium oxide nanoparticles, including morphology, size, absorbance, and surface charge. The multi-enzymatic properties, including catalase-like activity, superoxide-like dismutase activity, hydrogen peroxide (H_2_O_2_), and hydroxyl radical (·OH) scavenging abilities, were also comprehensively determined and compared, respectively. Finally, several biological properties, including intracellular ROS scavenging, modulation of macrophage phenotype, promotion of vascular endothelial cell migration, and angiogenesis, were been evaluated using a series of cellular assays.

## 2. Materials and Methods

### 2.1. Chemicals and Reagents

Cerium nitrate hexahydrate (Ce(NO_3_)_3_·6H_2_O), polyacrylic acid (PAA) 50% solution, ammonia (NH_3_·H_2_O), ferrous sulfate heptahydrate (II) (FeSO_4_·7H_2_O), N-(3-Dimethylaminopropyl)-N′-ehylcarbodiimide hydrochloride (EDC) and N-hydroxysuccinimide (NHS), and 4% paraformaldehyde fixative were purchased from Aladdin Reagent Co., Ltd. (Shanghai, China). c(RGDyk) was obtained from Hefei Bankpeptide Biotechnology Co., Ltd. (Hefei, China). MES buffer (pH = 5.50) and borate (BBS) buffer (pH = 8.00) were obtained from Beijing Leagene Biotechnology Co., Ltd. (Beijing, China). The 30% hydrogen peroxide (H_2_O_2_) was acquired from China National Pharmaceutical Group Co., Ltd. (Beijing, China). Salicylic acid was purchased from Maya Biotechnology Co., Ltd. (Shanghai, China). A reactive oxygen assay kit (DCFH-DA), fluorescent red dye, and SOD activity assay kit (WST-8 method) were obtained from Beyotime Biotechnology Co., Ltd. (Shanghai, China). CCK-8 cell proliferation detection kit, Hoechst 33342 staining solution, DMEM incomplete high-glucose culture solution, RPMI.1640 Incomplete Medium (1X), fetal bovine serum (FBS), trypsin EDTA digestion solution, and phosphate buffer PBS (1X) were purchased from Jiangsu KeyGEN Biotechnology Co., Ltd. (Nanjing, China). CD86-APC antibody and CD206-Alexa Fluor 488 antibody were purchased from Thermo Fisher Scientific Co., Ltd. (Shanghai, China). Matrigel Basement Membrane Matrix was obtained from Corning Life Sciences Co., Ltd. (Wujiang, China).

### 2.2. Synthesis and Characterizations of CeO_2_@PAA and CeO_2_@PAA@RGD

Synthesis: The CeO_2_@PAA was prepared according to the previously reported protocol developed by Asati et al. with minor modifications (Appendix A) [29]. Briefly, a total of 2.17 g (Ce(NO_3_)_3_·6H_2_O) was dissolved in 5 mL of deionized water at room temperature. The solution was then mixed with 10 mL of PAA (2.5 M). Then, the mixture was added dropwise into 30 mL of NH_3_·H_2_O under stirring at room temperature for 24 h. Subsequently, the CeO_2_@PAA solution was then concentrated using a 30 KDa ultrafiltration centrifuge tube to remove any free PAA.

The modification of the c(RGDyK) peptide (M = 619.69) on CeO_2_@PAA was based on the amide condensation reaction [38]. The obtained CeO_2_@PAA (30 mg, Ce element mass in CeO_2_ NP_S_) was dispersed in the 10 mL of MES buffer (pH = 5.50). EDC (30 mg) and NHS (20 mg) were added to the above solution and stirred for 30 min. Then, the mixture was purified using a 10 K ultrafiltration centrifuge tube to remove the extra EDC and NHS. Subsequently, 10 mg of c(RGDyK) peptides was dissolved in borate buffer (pH = 8.20) and added to the above reactants to react for 24 h under shaking conditions. After the reaction, the CeO_2_@PAA@RGD solution was obtained by centrifugation and ultrafiltration using a 10 KDa ultrafiltration tube.

Characterizations: The CeO_2_@PAA@RGD nanoparticles were characterized by a transmission electron microscope (JEM-200CX, JEOL, Tokyo, Japan) and dynamic light scattering (DLS, Malvern Nano-ZS). The UV–Vis absorbance spectra were detected by spectrophotometer (UV3600, Shimadzu, Kyoto, Japan). The chemical state of the surface of CeO_2_@PAA@RGD nanoparticles (percentage of Ce^3+^ and Ce^4+^) was evaluated by X-ray photoelectron spectroscopy (XPS, Thermo Escalab 250Xi, Waltham, MA, USA).

### 2.3. CAT Mimetic Activity Assay

The CAT activity of cerium oxide nanoparticles was measured by monitoring the generated O_2_ content from the decomposition of H_2_O_2_ using a dissolved oxygen electrode (JPSJ-605F, Lei ci, Shanghai, China) according to the previously reported literature [39,40]. Briefly, nanoparticles (500 µL, 10 mg/mL, calculated as Ce) were added into 8 mL of Tris-Cl buffer (pH = 7.48), and then 500 µL of H_2_O_2_ (30%) solution was added into the solution. The ability of CeO_2_@PAA@RGD to catalyze the decomposition of H_2_O_2_ to produce O_2_ was detected by the dissolved oxygen electrode in 30 min according to the manufacturer’s instruction [39].

### 2.4. SOD Mimetic Activity Assay

The SOD activity of cerium oxide nanoparticles was measured with an SOD assay kit. The amount of formazan produced by the (2-(2-methoxy-4-nitrophenyl)-3-(4-nitrophenyl)-5-(2,4-disulfophenyl)-2H-tetrazolium monosodium salt) (WST-8) with O_2_^−^ was directly correlated with the number of O_2_^−^. The substances with SOD enzyme activity can catalyze the disproportionation of O_2_^−^, thus decreasing the production of formazan dye. The quantitative determination of SOD mimetic activity was obtained by recording the absorbance of formazan at 450 nm. The concentration of nanoparticles used in the experiment was 10 mg/mL (calculated as Ce). The specific reaction system is shown in Appendix A. The percentage inhibition of O_2_^−^ was calculated by Equation (1).
Percentage inhibition = [(A blank control 1 − A blank control 2) − (A sample − A blank control 3)]/(A blank control 1 − A blank control 2)∗100%(1)

### 2.5. H_2_O_2_ Scavenging

The detection method of H_2_O_2_ refers to the previously reported literature using the Amplex-Red reagent assay to detect the absorbance at 570 nm [40]. Briefly, 5 μL of cerium oxide dispersions (10 mg/mL, Ce element mass in CeO_2_ NP_S_) was added into 100 μL of Tris-HCl buffer (pH = 7.50) and mixed with 10 μL of an H_2_O_2_ (120 mg/mL) solution. The reaction was performed for 10 min at room temperature. Then, 10 µL of Amplex Red (10 mM) reagent and 5 µL of horseradish peroxidase (400 μg/mL) were added and quickly transferred to a multifunctional enzyme labeler (Infinite M200, Männedorf, Switzerland to measure the absorbance at 570 nm.

### 2.6. Hydroxyl Radicals (·OH) Scavenging

The hydroxyl radicals (·OH) scavenging assay was performed according to the report by Zhou et al. [41]. ·OH was produced by the Fenton reaction: H_2_O_2_ + Fe^2+^ = ·OH + H_2_O + Fe^3+^. Salicylic acid can react with ·OH to produce 2,3-dihydroxybenzoic acid, which has a specific absorption at 510 nm. When an ·OH scavenger is added to the reaction system, the production of colored compounds would be reduced accordingly. All reagents were sequentially added to the 96-well plates according to Appendix A and reacted at 37 °C for 30 min. The absorbance at 510 nm was measured using a multifunctional enzyme standard (Infinite M200, Männedorf, Switzerland). The scavenging rate of ·OH (X) was calculated by Equation (2).
(2)$$X=\frac{A_{0}-\left(A_{x}-A_{x0}\right)}{A_{0}}*100$$

### 2.7. Cytotoxicity

A Cell Counting Kit-8 (CCK8) was used to evaluate the cytotoxicity of nanoparticles according to the previously reported method by Du et al. [42]. The cells (Raw264.7 or HUVEC) were seeded into 96-well plates with an equal density of 5 × 10^4^ cells and cultured in 100 μL of DMEM (Raw264.7) or RPMI.1640 (HUVEC) medium supplemented with 10% FBS prior to the assay, respectively. Then, cells were treated with a medium containing different concentration of nanoparticles (CeO_2_@PAA or CeO_2_@PAA@RGD) varying from 0–100 μg/mL (calculated based on the amount of Ce content in CeO_2_) and incubated for an additional 12 h or 24 h, respectively. Untreated cells were chosen as a control group. At the end of the treatment, the cells were rinsed with PBS three times and incubated in a serum-free medium containing 10% CCK8 for 2 h. Finally, the absorbance was measured at 450 nm using a multifunctional enzyme marker (Infinite M200, USA). All samples were measured in triplicate. Relative cell viability was calculated by Equation (3).
Relative viability = [(absorbance value of experimental group − absorbance value of blank group)/(absorbance value of control group − absorbance value of blank group)] ∗ 100(3)

### 2.8. Cellular Uptake

The cellular uptake of the different nanoparticles by HUVEC cells was examined according to the method previously reported by Yu et al. [43]. HUVEC cells were seeded in the laser confocal Petri dishes and incubated overnight. Then, the cells were treated with CeO_2_@PAA or CeO_2_@PAA@RGD containing RPMI-1640 culture medium for 4 h or 8 h, respectively. After rinsing with PBS buffer, the cells were fixed with 1 mL of 4% formaldehyde for 15 min at 4 °C condition, followed by another round of PBS washing. The cell nuclear staining was performed using Hoechst33342. The cells were imaged using a CLSM imaging system (TI2-S-HU, Nikon, Tokyo, Japan).

### 2.9. Intracellular ROS Scavenging in Raw264.7 Cells

The intracellular ROS-scavenging ability of different nanoparticles were compared. Briefly, Raw264.7 cells were seeded in the laser confocal Petri dishes and stimulated with H_2_O_2_ (100 μM) for 12 h to establish an oxidative stress cell model, followed by treatment with different nanoparticles for 24 h, including CeO_2_@PAA or CeO_2_@PAA@RGD (100 μg/mL, calculated based on the amount of Ce content). Cells without H_2_O_2_ treatment were used as a negative control group. After the treatment was completed, the cells were washed with PBS three times and stained with DCFH-DA (10 μM) for 30 min. Finally, intracellular fluorescence levels were observed and measured using a laser confocal microscope (TI2-S-HU, Japan, excitation wavelength: 488 nm, emission wavelength: 525 nm) and a Becton Dickinson FACScan analyzer (Franklin Lakes, NJ, USA).

### 2.10. Assessment of Macrophages Polarization In Vitro

The cellular expression of CD86 (a M1 macrophage marker) and CD206 (a M2 macrophage marker) were detected by flow cytometry (Becton Dickinson, Franklin Lakes, NJ, USA) to assess the extent of macrophage polarization according to the previous reported [44]. Briefly, Raw264.7 cells were seeded into the 6-well plate and cultured in DMEM medium supplemented with 10% FBS overnight, followed by being treated with LPS (2 μg/mL) for another 24 h. The cells treated with LPS were subsequently treated with CeO_2_@PAA@RGD or CeO_2_@PAA (100 μg/mL) for 24 h, respectively. At the end of the study, the cells were incubated with CD86-APC diluted with 1% BSA/PBS for 30 min at room temperature. Then, the cells were washed with PBS and were fixed in 4% paraformaldehyde for 15 min at room temperature, permeabilized with 1% (*v*/*v*) PBS/Trixton-100 for 5 min, followed by incubation with CD206-Alexa Fluor 488 diluted with 1% BSA/PBS for 30 min at room temperature. The value of the average fluorescence intensities of positive CD206 and CD86 was found using flow cytometry (Becton Dickinson, Franklin Lakes, NJ, USA).

### 2.11. Qrt-PCR Detection of Anti-Inflammatory and Proinflammatory Cytokines

The expression levels of proinflammatory markers (CD86, tumor necrosis factor-alpha (TNF-α) and interleukin-6 (IL-6)) in M1 cells and anti-inflammatory markers (CD206, vascular endothelial growth factor (VEGF), and interleukin-10 (IL-10)) in M2 cells were further analyzed by quantitative RT-PCR analysis. Raw264.7 cells were treated with the same method as 2.10 above. Total RNA was extracted using Trizol reagent and quantified by the fluorescence quantitative PCR instrument (Bio-rad CFX). The Cdna was then synthesized by reverse transcription according to the reaction system in Appendix A. The sequences of the primers are listed in Appendix A of the Supporting Information. The expression gene levels were normalized to GAPDH used as a housekeeping gene, and the relative levels were measured by the 2^−(ΔΔCT)^ method [45].

### 2.12. In Vitro Tube Formation of HUVEC

The HUVEC tube information was examined using Matrigel (Corning 356234, Nanjing, China) precoated in 96-well plates to assess the effect of the CeO_2_@PAA@RGD on the tube formation of HUVEC. The detailed procedure referred to that previously reported by Liu et al. [46]. HUVEC cells were cultured in a starving condition one day before the experiment. Then, HUVEC cells treated with different cerium oxide nanoparticles, including CeO_2_@PAA or CeO_2_@PAA@RGD (25 μg/Ml, calculated based on the amount of Ce content), were seeded on the polymerized Matrigel plates. After incubation for 2 h, 4 h, and 8 h, tube formation was observed using the microscope (CKX53, Tokyo, Japan), respectively. The node points and branches’ lengths were also calculated using ImageJ software.

### 2.13. Promotion of Cell Migration

Cell migration was recorded by microscopy at different time points to assess the effect of CeO_2_@PAA@RGD on HUVEC cell migration. According to the previously reported method [47], HUVEC cells were seeded into 6-well plates. After overnight incubation, a straight scratch was created with a pipette tip (200 μL), and the cells were gently washed with PBS three times. The fresh medium containing different cerium oxide nanoparticles, including CeO_2_@PAA or CeO_2_@PAA@RGD (25 μg/mL, calculated based on the amount of Ce content), was added. Images were taken at different time points with a microscope. The gap of the wound scratch was then determined using ImageJ software, and the percentage of wound healing was calculated according to Equation (4).
Healing rate = [(initial area − area at a time point)/initial area]*100 %(4)

### 2.14. Statistical Analysis

Statistical analysis of the experimental data was calculated using GraphPad and Origin software. The images were analyzed by ImageJ. One-way ANOVA was conducted during the statistical analysis; *p*-values represent significant differences. A *p* > 0.05 indicates no significant difference (ns) in experimental data. All * *p* < 0.05, ** *p* < 0.01, or *** *p* < 0.001 were considered statistically significant.

## 3. Results and Discussion 

### 3.1. In Vitro Physicochemical Characterization, Enzymatic Activity, and ROS-Scavenging Ability of CeO_2_@PAA@RGD

The CeO_2_@PAA nanoparticles were conjugated with RGD peptides to form CeO_2_@PAA@RGD. The DLS analyzer data showed that after the RGD modification, the hydrodynamic diameter of CeO_2_@PAA@RGD increased to 14.93 ± 1.52 nm from 6.84 ± 0.15 nm of CeO_2_@PAA (Figure 1a). The zeta potentials of CeO_2_@PAA and CeO_2_@PAA@RGD nanoparticles in a neutral aqueous solution were −23.80 ± 1.38 mV and −11.63 ± 1.41 mV, respectively (Figure 1b). The change in zeta potentials after conjugation was due to the RGD occupying part of the free carboxyl group on CeO_2_@PAA. TEM imaging indicated that CeO_2_@PAA@RGD presented a regular special shape with a uniform distribution and an average diameter of 3.35 ± 0.04 nm (Figure 1c), which was lower than the hydrodynamic size, due to the increased hydration layer caused by the PAA and RGD peptide modification on the particle surface. Meanwhile, the UV-Vis spectra showed that the CeO_2_@PAA@RGD had a characteristic absorbance peak similar to the CeO_2_@PAA at around 280 nm (Figure 1d), which is consistent with that previous reported by Lei et al. [48]. X-ray photoelectron spectroscopy (XPS) analyses were used to determine the surface Ce^3+^/Ce^4+^ ratio on the surface of the CeO_2_@PAA@RGD nanoparticles. Two main peaks, including Ce 3d 3/2 and Ce 3d 5/2, in the Ce 3d spectrum of CeO_2_@PAA@RGD are presented in Figure 1e. The Ce 3d spectrum was decomposed into six peaks [49], including four peaks of Ce^4+^ and two peaks of Ce^3+^ [41]. The peaks of Ce^3+^ are 881.90 and 900.60 eV, while the peaks of Ce^4+^ at 885.30, 904.00, 887.10, and 906.30 eV. The ratio of the total integrated area under the fitted peaks of Ce^3+^ (or Ce^4+^) to the total area of all peaks was used to calculate the relative concentration of the oxidized state of Ce^3+^ (or Ce^4+^) on the surface of cerium oxide nanoparticles, and it was found that the concentration of Ce^4+^ (63.05%) was higher than that of the Ce^3+^ (36.95%) oxidation state. Overall, these results implied the successful formation of CeO_2_@PAA@RGD.

We further determined the O_2_ generation ability via a time-dependent H_2_O_2_ decomposition assay to evaluate the catalase mimic enzyme (CAT) of CeO_2_@PAA@RGD. As shown in Appendix A, a large number of O_2_ bubbles were observed, and the solution showed a significant yellow color in the presence of nanoparticles. The color generated could be ascribed to the oxidation of Ce^3+^ to Ce^4+^ [40]. In contrast, almost no oxygen bubbles were observed in the PBS, and the solution appeared clear and colorless. The dynamic monitoring further showed that the oxygen concentration in the PBS group was almost unchanged for 30 min continuously. However, the dissolved O_2_ produced by the decomposition of H_2_O_2_ increased with the extended reaction time in the presence of nanoparticles (Figure 1f). The results indicated that CeO_2_@PAA@RGD had catalase-like catalytic activity. The RGD peptide modification did not affect its catalytic activity.

Superoxide dismutase (SOD) is a critical antioxidant enzyme in cells; it can catalyze the decomposition of O_2_^−^ into O_2_ and H_2_O_2_ [50]. The SOD mimetic activity of the CeO_2_@PAA@RGD was evaluated using the WST-8 method. Figure 1g indicates that compared to the control PBS (0.46%), the O_2_^·−^ inhibition rate of CeO_2_@PAA and CeO_2_@PAA@RGD reached 21.67% (*p* < 0.01) and 34.26% (*p* < 0.001), respectively. In addition, the RGD peptide modification further increased the inhibition rate of CeO_2_@PAA@RGD by 14.59% (*p* < 0.05) compared to that of CeO_2_@PAA (at the same Ce concentration). The enhanced SOD-mimicking activity of CeO_2_@PAA@RGD may be due to the improved surface affinity resulting from RGD peptide modification.

The ROS-scavenging activity of CeO_2_@PAA@RGD was further demonstrated. Two representative ROS, including H_2_O_2_ and ·OH, were selected for this assay. The results suggested that both nanoparticles were effective in scavenging H_2_O_2_ and ·OH compared to the control group. Moreover, there was no significant difference between CeO_2_@PAA and CeO_2_@PAA@RGD for H_2_O_2_ scavenging (Figure 1h), which proved that the surface modification of nanoparticles has almost no effect on H_2_O_2_ scavenging. In contrast, the inhibition of ·OH by CeO_2_@PAA@RGD was 12.72% higher than that of CeO_2_@PAA, indicated by the significantly decreased absorbance (*p* < 0.001) at 510 nm (Figure 1i), which demonstrated the effective removal of ·OH. These results suggest that the CeO_2_@PAA@RGD exhibits multienzyme mimetic activities and can significantly scavenge ROS.

### 3.2. Cytotoxicity Tests

Raw264.7 cells and HUVEC cells were employed to evaluate cytotoxicity upon interaction with CeO_2_@PAA and CeO_2_@PAA@RGD particles. Cell toxicity was determined using Cell Counting Kit-8 (CCK8) proliferation assay under different concentration of CeO_2_@PAA and CeO_2_@PAA@RGD (25, 50, and 100 μg/mL) treating for 12 h or 24 h, respectively. No notable toxicity was observed for both CeO_2_@PAA and CeO_2_@PAA@RGD on two types of cells under all tested concentrations during the different incubation period, suggesting the good biocompatibility of modified CeO_2_ nanoparticles (Figure 2a,b).

Moreover, we further assessed the RGD-mediated targeted uptake of CeO_2_@PAA@RGD by HUVEC cells with significant levels of integrin expression by confocal laser scanning microscopy (CLSM). CLSM images showed that iFluor-CeO_2_@PAA was not apparently taken up by HUVEC cells (Figure 2c). In contrast, iFluor-CeO_2_@PAA@RGD was notably internalized, as evidenced by the cytoplasmic distribution of red fluorescence noted. The cellular uptake level of CeO_2_@PAA@RGD was notably higher than CeO_2_@PAA, and the cellular uptake of CeO_2_@PAA@RGD had a time-dependent manner. The above results effectively illustrated the preferential uptake of CeO_2_@PAA@RGD by HUVEC cells via RGD-mediated endocytosis, which was consistent with the previous reports [35,51].

### 3.3. ROS-Scavenging Effects

The ROS-scavenging effects of CeO_2_@PAA@RGD on Raw264.7 cell were evaluated under a cellular oxidative microenvironment induced by 100 µm of H_2_O_2_. Intracellular ROS content was indicated via the ROS probe DCFH-DA (green color). As depicted in Figure 3b, H_2_O_2_ (100 μM) triggered a remarkably higher ROS in Raw264.7 than untreated control group (*p* < 0.001). Negligible fluorescence (*p* < 0.001) was observed in Raw264.7 cells treated with CeO_2_@PAA@RGD (100 μg/mL), which was comparable to the negative control (*p* > 0.05), suggesting the potent ROS-scavenging effect of CeO_2_@PAA@RGD. A similar significant decrease in fluorescence intensity was observed in the CeO_2_@PAA group compared to the H_2_O_2_ group (*p* < 0.001), but no significant difference was found compared to the CeO_2_@PAA@RGD group (*p* > 0.05). It was speculated that the comparable ROS-scavenging property of two tested cerium oxide nanoparticles resulted from the redox properties of Ce^3+^ and Ce^4+^ on the surface; the surface modifications had no significant effect on its redox properties. The effect of ROS scavenging was further confirmed by flow cytometry analysis (Figure 3c,d); CeO_2_@PAA and CeO_2_@PAA@RGD consistently showed significant ROS-scavenging levels in Raw264.7 cells, showing that the average fluorescence intensity of CeO_2_@PAA and CeO_2_@PAA@RGD treated cells was 2.87- and 3.89-fold lower than that of H_2_O_2_ (100 μM) (*p* < 0.001). The above results demonstrate that CeO_2_@PAA@RGD nanoparticles could effectively scavenge the intracellular ROS level and exhibit the antioxidant potential.

### 3.4. Anti-Inflammatory Effects

The potential anti-inflammatory abilities of CeO_2_@PAA@RGD were evaluated by macrophage phenotype transition and subsequent inflammatory cytokine detection. Firstly, CD86 (a common phenotypic marker of M1 macrophages) and CD206 (a common phenotypic marker of M2 macrophages) were detected for cells after different treatments, and the levels of inflammatory cytokines were assessed by flow cytometry after different treatments. The CD86 and CD206 expression levels are shown in Figure 4a. Raw264.7 cells treated with LPS (2 μg/mL) for 24 h produced 96.50% of CD86^+^ marker without notable CD206^+^ marker expression (Figure 4a). However, compared to the LPS group, the expression of CD86^+^ were significantly reduced in the cells treated with CeO_2_@PAA (2.57%) or CeO_2_@PAA@RGD (2.24%) groups. Moreover, a higher increment of CD206^+^ expression was observed in the cells treated with CeO_2_@PAA (45.60%) or CeO_2_@PAA@RGD (60.10%). Among them, the expression of CD206^+^ was 14.5% higher in the CeO_2_@PAA@RGD group than that in the CeO_2_@PAA group, suggesting that CeO_2_@PAA@RGD has a more potent ability to switch macrophage phenotype from M1 to M2 than CeO_2_@PAA. The inflammation-related cytokine expression was also tested by qRT-PCR to further show the anti-inflammatory effect of CeO_2_@PAA@RGD nanoparticles. As shown in Figure 4b, the expression of proinflammatory cytokines including CD86, TNF-α, and IL-6 were downregulated in either CeO_2_@PAA or CeO_2_@PAA@RGD groups compared to the LPS group. We also noted that the expression of anti-inflammatory cytokines such as CD206, VEGF, and IL-10 was increased accordingly compared to the LPS group (Figure 4c). These results demonstrate that CeO_2_@PAA@RGD has a promising anti-inflammatory effect.

### 3.5. Promotion of HUVEC Angiogenesis by CeO_2_@PAA@RGD

Tube formation assays of HUVEC were used to assess the CeO_2_@PAA@RGD effect on angiogenesis in vitro. The diagram shows the process of tube network formation (Figure 5a). The results showed that only a few tube networks were observed after incubating PBS for 8 h. Additionally, compared to the CeO_2_@PAA group, the CeO_2_@PAA@RGD induced more tube networks. With increasing time, the effect of promoting tube formation became more remarkable, and the number of node points and total tube length per field at 8 h were 2.14-fold and 1.50-fold higher than CeO_2_@PAA (Figure 5b,c), respectively. The CeO_2_@PAA@RGD showed stronger angiogenesis compared to the CeO_2_@PAA, which was due to the targeting property of RGD leading to greater cellular uptake of CeO_2_@PAA@RGD [52]. Collectively, these results suggested that the CeO_2_@PAA@RGD had a promoted angiogenesis of endothelial cells.

### 3.6. Promotion of Cell Migration

HUVEC cell migration is regarded as one of the critical steps in the angiogenesis process. We conducted a scratch wound healing assay to further evaluate the migration-promoting effect of CeO_2_@PAA@RGD on HUVEC. The images of scratch experiments for each group at different time points (0, 12, 24, 36 h) are displayed in Figure 6a, and it was observed that both the PBS group and CeO_2_@PAA-treated HUVEC cells did not fill the scratch area within 36 h. In contrast, the CeO_2_@PAA@RGD group presented the maximum wound recovery rate (97.95 ± 0.41%) after 36 h of incubation, which revealed almost a 2.72-fold and 1.40-fold increment compared to PBS (35.99 ± 2.81%) and CeO_2_@PAA (69.77 ± 0.78%), respectively (Figure 6b). These results suggested that the CeO_2_@PAA@RGD significantly promoted cell migration, consistent with the previous report [53].

RGD peptides are well known to bind preferentially to integrins αvβ3 [33], which are highly expressed in endothelial cells and play an important role in angiogenesis. Therefore, RGD peptide-modified nanoparticles are a promising approach to modulate the interaction between nanoparticles and endothelial cells. In this study, we found that CeO_2_@PAA@RGD significantly enhanced angiogenesis and cell migration in HUVEC cells. This may be attributed to the increased uptake of cerium oxide nanoparticles by endothelial cells after RGD modification of CeO_2_@PAA, thus further enhancing the interaction between nanoparticles and cells. It has been reported that low levels of intracellular ROS [54] and appropriate oxygen concentration [55] in the HUVEC cells are essential for anagenesis. The results of this paper demonstrate that CeO_2_@PAA@RGD has excellent CAT and SOD enzyme-like activities. It can effectively reduce the intracellular ROS level. CeO_2_@PAA@RGD has a greater potential in regulating intracellular oxygen concentration, which is mainly attributed to the fact that CeO_2_@PAA@RGD can catalyze the decomposition of H_2_O_2_ to produce oxygen. Based on the above hypothesis, it can effectively promote endothelial cell migration and angiogenesis.

## 4. Conclusions

In summary, CeO_2_@PAA@RGD nanoparticles maintained the excellent CAT-like and SOD-like enzymatic activities after the surface modifications. Due to the RGD modification, CeO_2_@PAA@RGD nanoparticles increased the cellular uptake by integrin-mediated endocytosis, thus enhancing the ROS-scavenging capability and exerting more potent antioxidant effects. In addition, the anti-inflammatory functions of CeO_2_@PAA@RGD were further confirmed by modulating the conversion of macrophages from M1 to M2 type, indicated by the upregulation of anti-inflammatory cytokines and downregulation of proinflammatory cytokines expression. Finally, the RGD modification promoted the uptake of CeO_2_@PAA@RGD by HUVEC cells, thus promoting the effects of angiogenesis and cell migration. Collectively, our study shed light on the development and application of CeO_2_@PAA@RGD nanoparticles for some inflammatory diseases, vascular-compromised diseases, diabetes, and tissue repair.

## Figures and Tables

**Figure 1 biomolecules-12-01277-f001:**
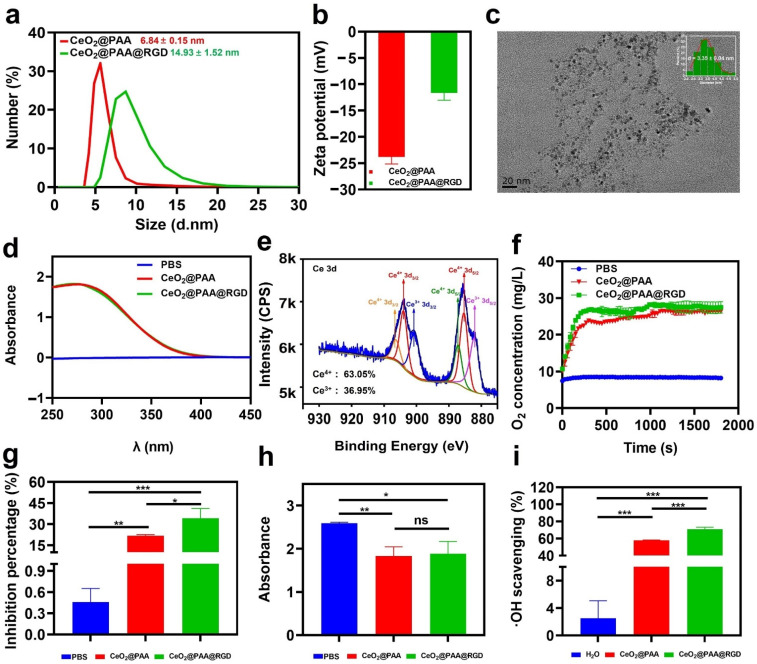
In vitro physicochemical characterizations, enzymatic activity, and ROS-scavenging ability of CeO_2_@PAA@RGD nanoparticles. (**a**) Hydrodynamic size distribution of CeO_2_@PAA (red line) and CeO_2_@PAA@RGD (green line) nanoparticles; (**b**) zeta potential analysis of CeO_2_@PAA (red) and CeO_2_@PAA@RGD (green) nanoparticles; (**c**) transmission electron microscopy (TEM) morphology characterization of CeO_2_@PAA@RGD nanoparticles. Scale bar = 20 nm (inset image: the TEM size distribution of CeO_2_@PAA@RGD nanoparticles); (**d**) UV-Vis spectra of CeO_2_@PAA and CeO_2_@PAA@RGD nanoparticles; (**e**) the X-ray photoelectron spectra of CeO_2_@PAA@RGD show the variation of spin-orbit double peaks of 3d 3/2 and 3d 5/2. The characteristic peaks of Ce^3+^ and Ce^4+^ are indicated by different colors; (**f**) the generation of oxygen was measured by a dissolved oxygen electrode to assess the catalase activity of CeO_2_@PAA@RGD; (**g**) the SOD-like activity of CeO_2_@PAA@RGD was assessed by calculating the percentage of superoxide anion inhibition; (**h**) the H_2_O_2_ scavenging ability of CeO_2_@PAA and CeO_2_@PAA@RGD was assessed by measuring the absorbance at 570 nm; (**i**) the scavenging percent of ·OH by CeO_2_@PAA and CeO_2_@PAA@RGD. Error bars = mean ± standard deviation (SD) (*n* = 3); ns > 0.05, * *p* < 0.05, ** *p* < 0.01, *** *p* < 0.001, using a one-way ANOVA (and nonparametric or mixed).

**Figure 2 biomolecules-12-01277-f002:**
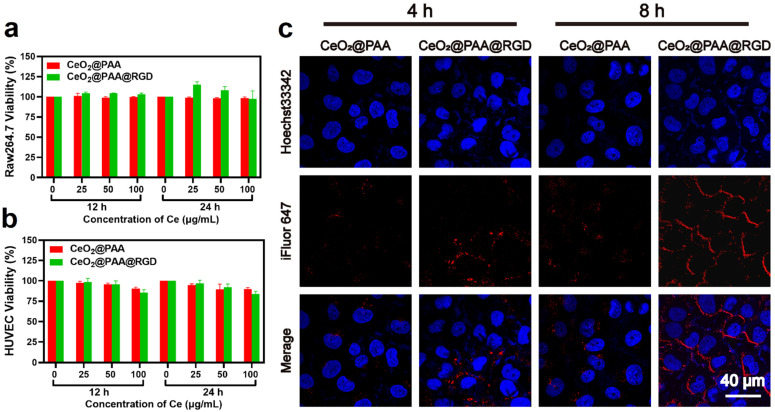
In vitro cytotoxicity and uptake assays. (**a**) Cytotoxicity of CeO_2_@PAA and CeO_2_@PAA@RGD on Raw264.7 cells for 12 h or 24 h incubation. Error bars = mean ± standard deviation (SD) (*n* = 3); (**b**) cytotoxicity of CeO_2_@PAA and CeO_2_@PAA@RGD on HUVEC cells for 12 h or 24 h incubation. Error bars = mean ± standard deviation (SD) (*n* = 3); (**c**) uptake of nanoparticles in HUVEC cells following 4 h or 8 h incubation by CLSM images to evaluate the targeting effect of CeO_2_@PAA@RGD. Cell nuclei were stained with Hoechst33342 with blue color and particles were stained with iFluor647 with red color. Scale bar = 40 µm.

**Figure 3 biomolecules-12-01277-f003:**
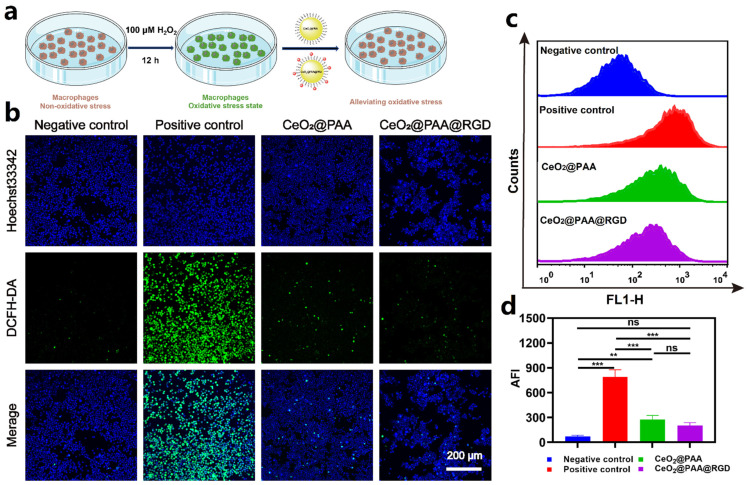
CeO_2_@PAA@RGD inhibited ROS production in Raw264.7 cells inducted by 100 µM H_2_O_2_ and showed antioxidative effects in activated Raw264.7. (**a**) Schematic illustration of in vitro ROS-scavenging capability of CeO_2_@PAA@RGD in Raw264.7 cells; (**b**) CLSM observation showing relative intracellular ROS levels after CeO_2_@PAA@RGD treated for 24 h. ROS were stained with DCFH-DA fluorescent probe (green) and nuclei were stained with Hoechst 33342 (blue). Scale bar = 200 µm; (**c**) histogram of ROS levels in Raw264.7 cells treated with CeO_2_@PAA@RGD nanoparticles for 24 h by flow cytometry analysis; (**d**) mean fluorescence intensity of intracellular ROS levels was calculated. Error bars = mean ± standard deviation (SD) (*n* = 3); ns > 0.05, ** *p* < 0.01, *** *p* < 0.001, using a one-way ANOVA (and nonparametric or mixed).

**Figure 4 biomolecules-12-01277-f004:**
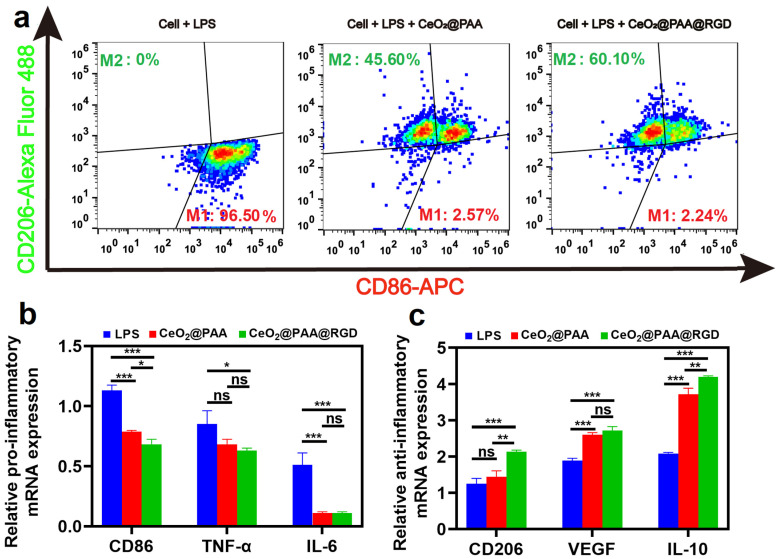
CeO_2_@PAA@RGD regulates the phenotype and function of macrophages by flow cytometry and qRT-PCR analysis. (**a**) Flow cytometry plots of CD86 and CD206 expression levels in macrophages after 24 h of CeO_2_@PAA or CeO_2_@PAA@RGD treated; (**b**) qRT-PCR was performed to detect the expression of proinflammatory factors CD86, TNF-α, and IL-6; (**c**) qRT-PCR was used to examine the expression of anti-inflammatory factors CD206, VEGF, and IL-10. Error bars = mean ± standard deviation (SD) (*n* = 3); ns > 0.05, * *p* < 0.05, ** *p* < 0.01, *** *p* < 0.001, using a one-way ANOVA (and nonparametric or mixed).

**Figure 5 biomolecules-12-01277-f005:**
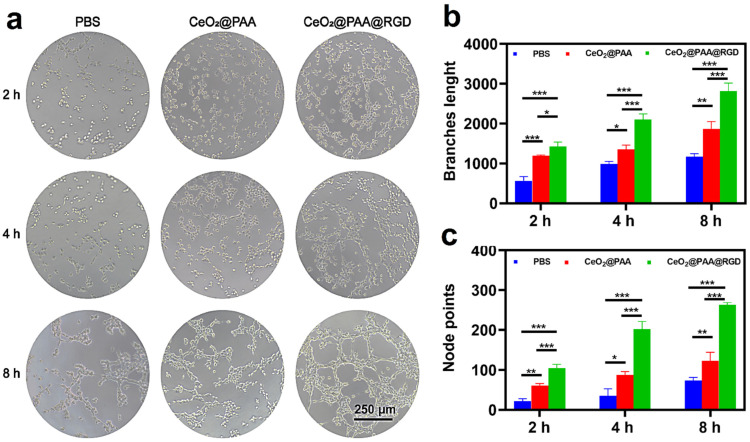
Angiogenic effect of HUVEC cells observed by microscopy. (**a**) Microscopic photomicrographs were used to observe the effect of angiogenesis in HUVEC cells after treatment with PBS, CeO_2_@PAA, and CeO_2_@PAA@RGD for 2 h, 4 h, and 8 h (scale bar = 250 μm); quantitative measurement of the branch length (**b**) and node points (**c**) (*n* = 3) after treatment of HUVEC cells with PBS, CeO_2_@PAA, and CeO_2_@PAA@RGD nanoparticles at 2 h, 4 h, and 8 h. Error bars = mean ± standard deviation (SD) (*n* = 3); * *p* < 0.05, ** *p* < 0.01, *** *p* < 0.001, using a one-way ANOVA (and nonparametric or mixed).

**Figure 6 biomolecules-12-01277-f006:**
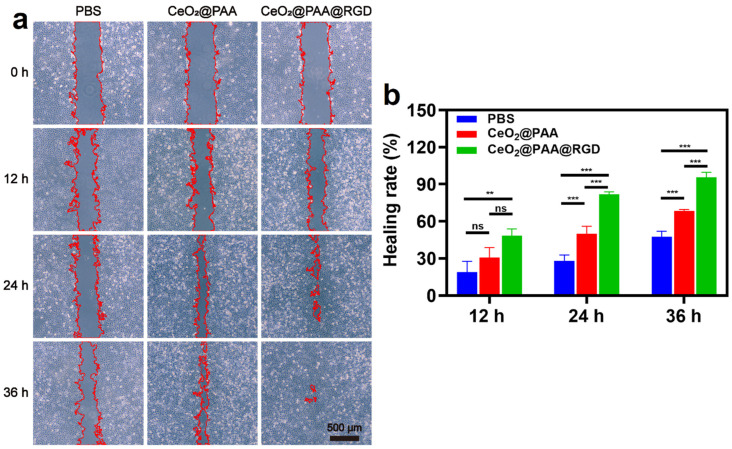
The migration of HUVEC was observed by microscopy in vitro. (**a**) The representative pictures of cell migration after treatment with PBS, CeO_2_@PAA, and CeO_2_@PAA@RGD nanoparticles for 0 h, 12 h, 24 h, and 36 h (red areas indicate scratch areas; scale bar = 500 µm); (**b**) quantification of scratch healing rate of HUVEC cells treated with PBS, CeO_2_@PAA, and CeO_2_@PAA@RGD nanoparticles. Error bars = mean ± standard deviation (SD) (*n* = 3); ns > 0.05, ** *p* < 0.01, *** *p* < 0.001, using a one-way ANOVA (and nonparametric or mixed).

## Data Availability

Not applicable.

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
