# Peer review of "Preparation, Characterization and Multiple Biological Properties of Peptide-Modified Cerium Oxide Nanoparticles"

_biomolecules, 2022, doi:10.3390/biom12091277_

Round 1
Reviewer 1 Report
This manuscript presents a novel approach to improving the biological properties of peptide-modified cerium oxide nanoparticles.
The manuscript is well documented and presents the results in chronological order, but the evaluation of the biological (enzymatic) properties is limited to the cell culture study. Some studies show that in vivo results may differ from those obtained in cell cultures or in vitro (for example: In Vitro and In Vivo Antioxidant Activity of the New Magnetic-Cerium Oxide Nanoconjugates. Nanomaterials. 2019; 9(11):1565. https .://doi.org/10.3390/nano9111565).
Revisions:
- Line 235 – (IL-10))
- Line 232-235 – Are they anti-inflammatory markers (CD86, TNFα, IL6) or pro-inflammatory? Similarly, CD206, VEGF, IL-10 - are they pro-inflammatory or anti-inflammatory?
- Line 428-434 - in this paragraph the cytokines with a pro-inflammatory role are cited as including CD86, TNFα, IL-6, and those with an anti-inflammatory role are CD206, VEGF, IL-10, the opposite of the way. are shown in lines 232-235.
I recommend publishing the article after the authors have clarified these issues.
Reviewer 2 Report
In this work, the authors prepared cerium oxide nanoparticles modified with poly (acrylic acid) (PAA) and then conjugated with an endothelium-targeting peptide glycine-arginine-aspartic acid (cRGD) to construct CeO2@PAA@RGD. Then the physico-chemical properties of fabricated nanoparticles have been characterized in detail. The “pseudo” multi-enzymatic properties as well as biological properties have been evaluated. I have carefully examined the manuscript and, while the topic is potentially interesting, several points need to be addressed.
1 – Page 2, Line 52-56: The authors mentioned that: “Among them, the smaller cerium oxide particles (less than 5 nm) have been demonstrated to exhibit greater enzyme activities [22,23]. However, the smaller cerium oxide nanoparticles are extremely unstable and prone to aggregate, thus resulting in the loss of enzymatic activities [24]. In addition, cerium oxide nanoparticles often suffer from insufficient cellular targeting property and then severe undesired toxicities [25]….” The statement is not absolutely correct.
The authors should revise this part:10.1039/d0ra08063b; 10.3390/biomedicines10050942; REF#4 doi:10.1021/acsami.0c08778.
2 – Page 3, Section 2.2: Characterization: Did authors determine the amount of RGD peptide on the surface of nanoparticles?
3 – Page 4, Section 2.4. SOD Mimetic Activity Assay. The authors should provide information about the concentration of nanoparticles used for SOD activity examination.
4 – Page 7, lines 277-280: The authors stated: “… The zeta potential measurements indicated the surface charge of CeO2@PAA@RGD increased to -11.63 ± 1.41 mV from -23.80 ± 1.38 mV of CeO2@PAA because of the shielding effect of RGD on the surface (Figure 1b). …” Authors should rewrite this sentence to avoid misinterpretation. It looks like that additional modification using RGD peptide led to a decrease in the colloidal stability of nanoparticles. (is this due to the presence of protein molecules on the surface?)
5 – Page 15: Please revise references carefully: Ref # 7 and 8 are the same.
